# Early Peri-Implant Bone Healing on Laser-Modified Surfaces with and without Hydroxyapatite Coating: An In Vivo Study

**DOI:** 10.3390/biology13070533

**Published:** 2024-07-17

**Authors:** Ana Flávia Piquera Santos, Rodrigo Capalbo da Silva, Henrique Hadad, Laís Kawamata de Jesus, Maísa Pereira-Silva, Heloisa Helena Nímia, Sandra Helena Penha Oliveira, Antônio Carlos Guastaldi, Thallita Pereira Queiroz, Pier Paolo Poli, Debora de Barros Barbosa, André Luis da Silva Fabris, Idelmo Rangel Garcia Júnior, Reinhard Gruber, Francisley Ávila Souza

**Affiliations:** 1Department of Diagnosis and Surgery, School of Dentistry, São Paulo State University (UNESP), Araçatuba 16015-050, SP, Brazil; h.hadad@unesp.br (H.H.); lais.kawamata@unesp.br (L.K.d.J.); maisa.silva@unesp.br (M.P.-S.); andre.fabris@hotmail.com (A.L.d.S.F.); idelmo.rangel@unesp.br (I.R.G.J.); 2Department of Dental Materials and Prosthetics, School of Dentistry, São Paulo State University (UNESP), Araçatuba 16015-050, SP, Brazil; rodrigo.capalbo@unesp.br (R.C.d.S.); h.nimia@unesp.br or helonimia@pucpcaldas.br (H.H.N.); debora.b.barbosa@unesp.br (D.d.B.B.); 3Health Sciences Institute, Pontificiae University Catholic of Minas Gerais—PUC-Minas, Poços de Caldas 37714-620, MG, Brazil; 4Department of Basic Sciences, School of Dentistry, São Paulo State University (UNESP), Araçatuba 16018-805, SP, Brazil; sandra.hp.oliveira@unesp.br; 5Department of Analytical, Physical-Chemistry and Inorganic Chemistry, Institute of Chemistry, São Paulo State University (UNESP), Araraquara 14800-900, SP, Brazil; ac.guastaldi@unesp.br; 6Department of Health Science, University of Araraquara-UNIARA, Araraquara 14801-340, SP, Brazil; thaqueiroz@hotmail.com; 7Maxillofacial Surgery and Odontostomatology Unit, Fondazione IRCSS Cà Granda Maggiore Policlinico Hospital, University of Milan, 20122 Milan, Italy; pierpaolo.poli@unimi.it; 8Department of Oral Biology, Medical University of Vienna, 1090 Vienna, Austria; reinhard.gruber@meduniwien.ac.at

**Keywords:** dental implants, healing, dental etching, YAG lasers, osseointegration

## Abstract

**Simple Summary:**

This research investigated the impact of different surface modifications on the osseointegration of dental implants. The study conducted experiments on rabbit tibiae comparing conventional machined surfaces (MSs) with those modified using a laser beam (LSs) or a laser beam incorporating hydroxyapatite (HA) by means of biomimetic methods without thermic treatment (LHSs). Utilizing scanning electron microscopy coupled with energy-dispersive X-ray spectrometry (SEM/EDX) and fluorochrome labeling techniques, the study evaluated bone formation over a four-week period. The results demonstrated significantly improved bone contact and newly formed bone area with LSs and LHSs compared to MSs. These findings suggest that surface modifications, particularly LS modifications and LHS modifications, enhance osseointegration, potentially leading to an enhanced longevity and performance of dental implants in clinical settings.

**Abstract:**

(1) Objective: The aim of this study was to assess the biological behavior of bone tissue on a machined surface (MS) and modifications made by a laser beam (LS) and by a laser beam incorporated with hydroxyapatite (HA) using a biomimetic method without thermic treatment (LHS). (2) Methods: Scanning electron microscopy coupled with energy-dispersive X-ray spectrometry (SEM/EDX) was performed before and after installation in the rabbit tibiae. A total of 20 Albinus rabbits randomly received 30 implants of 3.75 × 10 mm in the right and left tibias, with two implants on each surface in each tibia. In the animals belonging to the 4-week euthanasia period group, intramuscular application of the fluorochromes calcein and alizarin was performed. In implants placed mesially in the tibiofemoral joint, biomechanical analysis was performed by means of a removal torque (N/cm). The tibias with the implants located distally to the joint were submitted for analysis by confocal laser microscopy (mineral apposition rate) and for histometric analysis by bone contact implant (%BIC) and newly formed bone area (%NBA). (3) Results: The SEM showed differences between the surfaces. The biomechanical analysis revealed significant differences in removal torque values between the MSs and LHSs over a 2-week period. Over a 4-week period, both the LSs and LHSs demonstrated removal torque values statistically higher than the MSs. BIC of the LHS implants were statistically superior to MS at the 2-week period and LHS and LS surfaces were statistically superior to MS at the 4-week period. Statistical analysis of the NBA of the implants showed difference between the LHS and MS in the period of 2 weeks. (4) Conclusions: The modifications of the LSs and LHSs provided important physicochemical modifications that favored the deposition of bone tissue on the surface of the implants.

## 1. Introduction

Dental implants have achieved notable success and long-term stability, which is largely attributed to advances in implant surface modifications that improve the bone–implant interface [1,2,3]. These modifications play a key role in ensuring a predictable, safe and durable connection between the implant and the bone, a process known as osseointegration- initially defined by Bränemark [4] as the structural, direct and functional connection between organized bone and the implant surface at a microscopic level, allowing the implant to resist masticatory forces [5]. Despite initial success with commercially pure titanium (cp-Ti) implants in good quality bone, the success rate in more challenging conditions, such as compromised bone, has been less promising [6,7].

To address these challenges, researchers have focused on improving implant surfaces to improve osseointegration, especially in suboptimal bone conditions [8]. Various surface modification techniques have been developed to optimize bone tissue deposition and implant success. These include mechanical methods such as sandblasting, acid etching and plasma spraying; chemical methods such as hydroxyapatite (HA) coatings; and biological methods involving the incorporation of biological agents [9,10,11,12,13,14,15]. These modifications improve protein adsorption, cellular behavior and overall biological responses, leading to higher removal torque values [9,16,17,18,19,20] and increased bone-to-implant contact [21,22,23].

Among the innovative surface modification techniques, laser beam ablation stands out for its precision, clean processing and high degree of purity, and can be performed in a controlled and reproducible manner [9,24,25,26]. Using lasers made of a mixture of gasses such as carbon dioxide (CO_2_) and yttrium and aluminum garnet lasers doped with ytterbium (Yb:YAG laser), this method allows controlled and reproducible surface modifications without the need for chemical agents that can degrade over time [10,11,27] in addition to the fact that Yb:YAG laser has sufficient and adequate energy to modify the surface of the implant [9,10,11,28,29]. Studies have shown that implants modified with a Yb:YAG laser have better bone-to-implant contact and higher removal torque values compared to machined surfaces [9,30,31,32,33].

Furthermore, the deposition of biomaterials on a previously modified surface has been an innovative trend. The most common and first employed was the HA plasma spray technique [34]. However, this technique has considerable disadvantages such as the need for high temperatures to perform the coating, leading to changes in the structure of HA, excessive porosity, solubilization and poor adhesion of HA to the implant surface, which can promote cracks and fractures in the coating [35]. To address these challenges, researchers have explored new deposition methods, including the biomimetic approach. This method involves immersing the substrate to be coated in a synthetic solution known as Simulated Body Fluid (SBF), which mimics the chemical composition, pH and temperature of human blood plasma. This approach has shown promising results in enhancing the bioactivity and integration of implants [36,37,38,39].

The discovery and improvement of biomimetic techniques allowed previously modified implant surfaces to be covered with different phases of hydroxyapatite or other biomaterials with osteoconductive capacity [9,10,40,41]. In view of the above advances, this study aims to evaluate the biological and mechanical behavior of bone tissue around implants with machined surfaces (MSs), laser modified surfaces (LSs) and laser modified surfaces with the incorporation of HA through the biomimetic method (LHSs) in the tibias of rabbits, under physiological conditions and during the initial phases of peri-implant bone healing.

## 2. Materials and Methods

### 2.1. Surfaces

Thirty commercially pure titanium implants (cp-Ti), grade IV with external hexagon connection, and dimensions of 3.75 × 10 mm (Titanium Fix, AS Technology, São José dos Campos, São Paulo, Brazil) with 3 different types of surfaces, were used. The machined surface (MS) was kindly donated by Titanium Fix, AS Technology, São José dos Campos, SP, Brazil. Twenty of the MS implants were taken to the Institute of Chemistry, São Paulo State University (UNESP), Araraquara, São Paulo, Brazil, to carry surface modifications by laser beam (LS) and by laser beam followed by deposition of HA using a biomimetic method without thermic treatment (LHS).

#### 2.1.1. Laser Beam

The samples were fixed in a rotating lathe under the pulsed Yb: 20 W laser equipment (Pulsed Ytterbium Fiber Laser, OmniMark System 20F, Ominitek Tecnologia Ltd., São Paulo, Brazil), with the parameters of 140 mJ nominal power supply, pulse frequency of 20 KHz, wavelength 1064 nm and cut-off length 10 µm. The laser beam was projected over the entire surface of the samples for 90 min at room temperature [10,12].

#### 2.1.2. Laser Beam Followed by Incorporation of Hydroxyapatite Biomimetic Method

After surface irradiation by laser beam, the samples were immersed in 50 mL of NaOH solution (5.0 Mol·L^−1^) in an oven for a period of 24 h at 60 °C for surface activation, and a layer of sodium titanate formed. After immersion in an alkaline solution, the samples were kept in the oven for 3 h to dry the surface and subsequently immersed in simulated body fluid (SBF) [38] which simulates body fluids by presenting an ionic composition and pH like those of blood plasma. The samples remained immersed in SBF for a period of 4 days, in an oven at 37 °C and at pH 7.25, to obtain a mixture composed of HA. The solution was changed every 24 h to maintain the number of ions.

### 2.2. Topographic Characterization of the Implants

#### 2.2.1. Scanning Electron Microscopy Analysis Coupled to the X-ray Dispersive-Energy Spectrometry System—SEM-EDX

The topography of the surface of the samples was analyzed using a scanning electron microscope (JEOL, model JSM-7500F, Tokyo, Japan, equipped with an EDX microanalysis detector of the Inca X-act model, Oxford, UK), coupled with dispersive-energy spectrometry X-ray (EDX), for semi-quantitative analysis of the chemical composition of surfaces.

#### 2.2.2. Contact Angle Measurements

The wettability of the samples was measured at room temperature, with a relative humidity of 75%, using equipment (Contact Angle System, video-based Dataphysics, model OCA-15) for the analysis of the contact angle. The measurement of each sample was repeated 3 times to obtain the average value of the contact angle (θ) of the different surfaces.

### 2.3. Animals and Ethics Committee

This study was approved by the Ethics Committee on Animal Experimentation (CEUA), established by the Brazilian College of Animal Experimentation (COBEA), of São Paulo State University (UNESP), School of Dentistry, Araçatuba, Brazil (#0554-2018), and the experiment was conducted in accordance with the relevant guidelines and legislation of the Animal Research such as the reporting In Vivo Experiments (ARRIVE) guidelines [42]. A total of 20 adult male rabbits, Albinus variation, provided by the Animal Facility of the São Paulo State University (UNESP), Medical School, Botucatu, at approximately 5 months old, weighing around 3.1 to 4.3 kg, were used and received 2 implants of 3.75 × 10 mm of each surface (AS Technology, TitaniumFix, São Paulo, Brazil), randomly installed in both tibias. After the surgical procedure, the animals were divided into periods of euthanasia of 2 and 4 weeks, comprising 5 animals per period and 10 implants on each surface in each period. Five implants were used in biomechanical and SEM-EDX analysis and five implants in confocal microscopy and histometric analysis.

Following methodologies of previously published studies [10,11,12,13,40,41], the animals were kept in preoperative fasting for 8 h before the surgical procedure. Before any surgical procedure, the animals were anesthetized with intramuscular infiltration of ketamine hydrochloride (50 mg/kg, Vetaset, Fort Dodge, Ltd., Campinas, São Paulo, Brazil) and xylazine (5 mg/kg, Dopaser, Ltd., Osasco, São Paulo, Brazil). Then, trichotomy and antisepsis were performed with Polyvinyl Pyrrolidone Iodine Degermant (PVP-I 10%, Riodeine Degermant, Rioquímica, São José do Rio Preto, São Paulo, Brazil). Then, local anesthetization was performed with mepivacaine hydrochloride (0.3 mL/kg, Scandicaine 2% with adrenaline 1:100,000, Septodont, France).

The surgical procedure involved a dermo-periosteal incision on the anterior border of the tibia, and the flap was displaced to expose the lateral face of the tibia. Two osteotomies were performed, and the implant recipient site was prepared using a spear-shaped drill, as well as 2.0 mm, 2.0/3.0 mm pilot and 3.0 mm helical cutters. The implants were installed using an internal torque wrench for an external hexagon (AS Technology, TitaniumFix, São Paulo, Brazil) and a cover screw was placed to protect the ridge module. Finally, the soft tissues were repositioned and sutured in layers using absorbable Polygalactin 910 thread (Vicryl 5.0, Ethicon, Johnson, São José dos Campos, São Paulo, Brazil) in the muscle and 5.0 nylon thread (Ethicon, Johnson, São José dos Campos, São Paulo, Brazil) in the cutaneous plane.

In the postoperative period, the animals received an intramuscular injection of pentabiotic (0.1 mL/kg, Fort Dodge Saúde Animal Ltd., Campinas, SP, Brazil) immediately following surgery. Additionally, a single daily dose of sodium dipyrone (1 mg/kg/day, Ariston Indústrias Químicas e Farmacêuticas Ltd., Cotia, SP, Brazil) was administered for 5 days.

### 2.4. Application of Fluorochromes

The application of fluorochromes followed the methodology used in previously published studies [43]. For the analysis of epifluorescence, 20 mg/kg of calcein was administered intramuscularly within 15 days of implant placement. After 25 days of implant installation, 30 mg/kg of alizarin red was administered to each animal.

### 2.5. Euthanasia and Material Collection

Sedation was performed following the same protocol as the surgical procedure. After the anesthetic was administered, the implant located more proximally to the tibiofemoral joint was assessed with biomechanical analysis by removal torque measurements. After analysis, the animals were euthanized by anesthetic overdose.

After euthanasia, the right and left tibias were removed for subsequent analysis. The implants removed by removal torque were separated for SEM/EDX and the implants located more distally to the tibiofemoral joint were used for confocal laser microscopy and histometric analysis.

### 2.6. Biomechanical Analysis

The internal torque implant assembler was installed, coupled to the ratchet extension wrench (AS Technology, Titanium Fix, São Paulo, Brazil). The implants were removed using a digital torquemeter (Data Tork CEM 3, Tohnichi Mfg. Co., Ltd., Tokyo, Japan), and the value required for implant removal was measured in N/cm. The 5 values of the 3 surfaces in each period of euthanasia were recorded and tabulated for statistical analysis.

### 2.7. Fluorochrome Analysis (Active Mineralized Surface)

The right and left tibias, containing the implants located more distally to the tibiofemoral joint, were placed in neutral buffered formalin at 10% for a period of 72 h. Then, these pieces were dehydrated in increasing concentrations of alcohol (60–100% ethanol) and subsequently encapsulated in light-cured resin (Technovit 7200 VLC, Kultzer Heraeus GmbH & Co., Wehrheim, Germany) for further processing on the Exakt system. The cutting and wear of the parts were made in the mesial–distal plane using a cutting system (Exakt^®^ Cutting System, Apparatebau, GmbH, Hamburg, Germany) to obtain about 50 μm thick sections.

The slides were captured by the Leica CTR 4000 CS SPE confocal laser microscope (Leica Microsystems, Heidelberg, Germany), using a 10× magnification (original increase 100) at the Center for Confocal Laser Microscopy at the São Paulo State University (UNESP), School of Dentistry, Araraquara, São Paulo, Brazil. For the quantification of values in the images obtained with the microscope, the ImageJ 1.53a image analyzer software (Java 1.8.0_172, US National Institutes of Health, Bethesda, MD, USA) was used. To measure the area of fluorochromes (calcein/red alizarin), the “free hands” tool (μm^2^) was used with a green fluorescent color (calcein) and a fluorescent red (alizarin). For the analysis of mineral bone apposition (MAR), 5 measurements were taken extending from the external margin of calcein towards the external margin of alizarin, and the value obtained was then divided by 10, which represents the interval of days between injections of the two fluorochromes analyzed [44].

### 2.8. Histometric Analysis

All slides obtained from the 20 animals containing implants in mineralized tissue from the 4 experimental groups and euthanized after 2 or 4 weeks were stained with Stevenel’s blue and acid fuchsin. The images were analyzed in an optical microscope (DM750, Leica Microsystems Vertrieb GmbH, Munich, Germany) (Diastar, Leica Reichert & Jung products, Wetzlar, Germany) and captured using a digital camera (Leica Microsystems, ICC50E Camera Module, Wetzlar, Germany) with a resolution of 1.3 megapixels coupled to the light microscope.

Histometric analyses were performed using the image analysis software ImageJ version 1.53a (Java 1.8.0_172, US National Institutes of Health, Bethesda, MD, USA). The percentage of bone implant contact (%BIC) and percentage of new bone area present (%NBA) between the 3 most coronal turns, located in the cortical bone on each side of the implant, were calculated as percentages [12,13].

### 2.9. Statistical Analysis

The values obtained in the biomechanics analysis, MAR and histometric analysis (BIC and NBA) were tabulated and the Kolmogorov–Smirnov test was used to verify the homogeneity of distribution. Afterwards, they were tested by Two-way ANOVA and Tukey’s multiple comparisons test.

## 3. Results

### 3.1. SEM/EDX

To determine the roughness pattern on each surface, SEM analysis was performed. The MSs presented a relatively smooth morphology, without porosity characteristics, which is indicative of a mechanically processed surface with minimal surface roughness and texture. The EDS spectrum revealed only titanium (Ti) peaks (Figure 1a). The LSs exhibited a homogeneous structure characterized by small closely packed particles, indicating a modified surface with increased surface area due to laser modification. The EDS spectrum shows dominant peaks for oxygen (O) and titanium (Ti) (Figure 1b). The presence of oxygen is likely due to the formation of titanium oxide (TiO_2_) as a result of laser modification, which typically induces oxidation on the titanium surface, increasing its chemical reactivity and improving surface properties for various applications.

The LHSs revealed a highly porous and rough surface morphology with numerous cavities and irregular features. This indicates a significantly altered surface texture due to laser ablation, further improved by the application of the HA coating. The corresponding EDS spectrum shows several peaks, including prominent peaks for oxygen (O), calcium (Ca) and phosphorus (P), as well as peaks for titanium (Ti), sodium (Na), magnesium (Mg), potassium (K), carbon (C) and chlorine (Cl). This composition suggests the successful deposition of hydroxyapatite, a calcium phosphate-based material known for its bioactive properties and compatibility with biological tissues (Figure 1c).

### 3.2. Contact Angle Measurements

To evaluate the wettability of the surfaces, measurements of the contact angle were performed. The images obtained during the measurement are shown in Figure 2 and the values of the contact angles of the surfaces are shown in Table 1. The MS had an average contact angle of approximately 60° (Figure 2a). Although this value is below the 90° threshold for hydrophobicity, it suggests that the MS is relatively close to hydrophobic in nature, indicating limited wettability.

In contrast, the LS and LHS exhibited contact angles of 0° (Figure 2b and 2c, respectively), indicative of superhydrophilic properties. This behavior is characterized by the immediate spreading of the water droplet upon contact, making it difficult to measure a distinct contact angle. The instantaneous spreading confirms the highly wettable nature of these surfaces, distinguishing them from the MS surface.

### 3.3. SEM/EDX of the Removed Implants

To demonstrate the presence of bone tissue, as well as to evaluate the chemical composition of implants removed by reverse torque, SEM/EDX of the surfaces was performed. The smoother morphology of the MS, with minimal surface roughness and texture as observed in Figure 1a, correlates with the almost negligible bone coverage observed in the 2- and 4-week periods (Figure 3A,D). On the other hand, the increase in surface area and the presence of titanium oxide on the LS and HA in LHS evidenced by the homogeneous structure with small particles (Figure 1b,c), correspond to the greater deposition of bone tissue observed at 2 weeks (Figure 3B,C) and 4 weeks (Figure 3E,F). This indicates an increase in bone–implant contact on the LS and LHS compared to the MS.

EDX analysis of the MS over the 2-week period (Figure 3a) showed peaks of Ti and O. In contrast, the LS (Figure 3b) showed peaks of O, Ti, Ca, P, and Na, while the LHS (Figure 3c) showed O, C, Ti, Ca, P and Na peaks. During the 4-week period, the EDX of the MS (Figure 3d) again showed peaks of Ti, O and Na. The LS (Figure 3e) exhibited higher peaks of O, Ti, C, Ca, P and Na, and the LHS (Figure 3f) exhibited peaks of O, C, Ca, Ti, P and Na. The higher values of oxygen, carbon, calcium and phosphate on the LS and LHS surfaces indicate that a greater amount of bone tissue was deposited on these surfaces at intervals of 2 and 4 weeks compared to the MS surface. This is further supported when compared to the EDX of implants before installation, where we predominantly observed Ti peaks in MS (Figure 1a), Ti and O peaks in LS (Figure 1b) and O, Ca P, Ti, Na, Mg, K, C and Cl peaks in the LHS due to the addition of HA (Figure 1c).

### 3.4. Biomechanical Analysis

The clinical evaluation showed no signs of infection, and no complete or partial tibial bone fracture was observed in any of the animals. No implants were lost, and all were stable and without marginal bone loss. In the MS group, the mean torque values for implant removal were 15 and 19.4 N/cm in periods of 2 and 4 weeks, respectively. In the LS group, the means found were 26.6 and 34.8 N/cm in the two periods of analysis. On the LHS, the mean torque values for removal were 30.2 and 35 N/cm. A significant difference was observed between the LHS and the MS in the 2-weeks period. In the 4-weeks period, a significant difference was observed between the LS and LHS when compared to the MS, as indicated in Figure 4.

### 3.5. Fluorochrome Analysis

To qualitatively evaluate the presence of the red fluorochromes alizarin and calcein, confocal laser microscopy was used. Red fluorescence indicates newly mineralized bone stained with alizarin red, while green fluorescence represents actively mineralized bone or older bone matrix stained with calcein. The combined images show the overlap of these fluorochromes, providing information about the bone remodeling process. MS implants showed moderate new bone formation, mainly around the peri-implant region (Figure 5a). Calcein staining showed a more dispersed pattern, suggesting ongoing bone remodeling and integration (Figure 5b). The yellow regions correspond to areas where new and old bone coexist, indicating a mixture of mature and actively forming bone (Figure 5c).

For the LS implants, alizarin red highlighted new, substantial bone formation that was more evenly distributed compared to the machined surfaces (Figure 5d). Calcein fluorescence was more pronounced, indicating greater mineralization activity, and the overlap of new and old bone suggests osseointegration and remodeling due to laser modification (Figure 5e,f). And, for the LHS implants, the intense alizarin red staining indicates robust, abundant, and evenly distributed new bone formation (Figure 5g). Calcein highlighted a dense and uniform mineralization pattern, suggesting a highly active bone remodeling process (Figure 5h). The combined image indicates effective osseointegration, with the hydroxyapatite coating further promoting bone formation and stabilization around the implant (Figure 5i).

To determine the MAR in the images obtained by confocal laser microscopy, quantification was performed using 5 measurements calculated by the values from the outer margin of red alizarin to the outer margin of calcein, divided by 10 and expressed as a percentage. The values were submitted to the normality test, and then, the two-way ANOVA was used with Tukey’s for multiple comparisons test. The comparison between the groups showed that a statistically significant difference was found between the LHSs and MSs, and between the LSs and MSs (Figure 6), showing a greater amount of newly formed bone tissue on the experimentally modified surfaces.

### 3.6. Qualitative Histological Analysis

Histological analysis of calcified sections obtained using the Exakt system was conducted with Stevenel’s blue and acid fuchsin staining. In the 2-week period (Figure 7, upper line), the MS group exhibited a reduced amount of trabecular bone in the initial threads of the implant within the cortical region, characterized by immature connective tissue, stained blue. The collagen fibers appeared less organized and less mature, and in some areas, a distinct separation between pre-existing and newly formed bone was observed (white arrows). In contrast, the LSs and LHSs in the cortical region showed regular and more homogeneous collagen fibers with concentric lamellae, indicative of ongoing remodeling, maturation and new tissue formation. Additionally, these surfaces demonstrated a significant amount of interface contact with the bone tissue.

By the 4-week period (Figure 7, lower line), the MS displayed a more mature bone pattern with an increased interface contact with bone tissue, although some areas of immature connective tissue persisted between the bone trabeculae, along with regions of preexisting bone that lacked remodeling (green stars). For the LS and LHS, there was enhanced tissue maturation and a significant contact interface with the bone tissue was maintained, reflecting continued integration and stability.

### 3.7. Histometric Analysis

Finally, to quantify the presence of bone tissue on the implants present in the calcified sections, the bone implant contact (%BIC) was measured, as well as the area of newly formed bone tissue (%NBA), both expressed as a percentage.

For bone-implant contact (%BIC), significant differences between surface treatments were observed within 2 weeks. MS had the lowest BIC values, averaging below 40%. In contrast, LS and LHS showed BIC values around 60%, suggesting early osseointegration. While there were statistical differences when compared to MS, no significant difference was observed between LS and LHS.

A similar pattern was noted during the 4-week period. Although the MS showed an increase in the BIC values, approaching 40%, there were still statistically significant differences between the MS and the treated surfaces. The LS and LHS did not exhibit a significant increase compared to the 2-week period, and no difference was observed between the two treated surfaces. These results suggest that the initial improvements in osseointegration provided by laser modification and HA deposition were sustained over the 4-week period (Figure 8).

For new bone area (%NBA), during the 2-week period, significant differences were observed only between the MS and LHS. The MS exhibited the smallest values of NBA, with an average of about 65%. The LS had NBA around 80%, while LHS showed the highest values, close to 85%. This indicates that the surface modifications were effective in accelerating bone formation around the implants in early stages.

During the 4-week period, the MS showed an increase in the NBA, reaching nearly 70%, but it remained the lowest among the three surfaces analyzed. Both the LS and LHS exhibited a decrease when compared to the 2-week period, with values around 75%. Although there was no statistically significant difference between the treated surfaces, it was evident that the LS and LHS had higher values compared to the MS (Figure 9).

## 4. Discussion

Optimizing implant surfaces has been a focal point in dental research due to its critical role in promoting osseointegration [14]. Previous studies have consistently demonstrated that surface topography significantly influences the biological response to implants by affecting the rate and quality of bone formation around the implant [3,11,40,41,45,46]. Modifying surface morphology through techniques such as laser modification and biomaterial coatings can improve initial bone–implant contact, leading to better clinical outcomes. The present study builds on these findings by investigating the impact of laser-treated surfaces, with and without an HA coating, on peri-implant bone formation. The results were compared with an MS to evaluate the effectiveness of these modifications. The use of scanning electron microscopy (SEM) and quantitative analysis of bone–implant contact (%BIC) provided a comprehensive assessment of surface topography and its correlation with bone formation. This study’s focus on the initial periods of bone healing (2 and 4 weeks) provides valuable information about the early stages of osseointegration, which are crucial to the long-term success of dental implants.

SEM analysis provided a detailed examination of the surface topographies. The MS exhibited a relatively smooth morphology, typical of mechanically processed surfaces with minimal roughness. This finding is in line with previous studies associating smoother surfaces with limited bone–implant integration due to there being a reduced surface area for cell attachment [45]. The relatively low wettability of MS, indicated by an average contact angle of approximately 60°, suggests limited surface interaction with biological fluids, which is a disadvantage in early bone formation [47]. This observation is consistent with the findings of Elias et al. [48], who reported that smoother surfaces tend to have a lower wettability, negatively impacting cell adhesion. In contrast, the LS demonstrated a homogeneous granular structure. This microtopography, characterized by small, closely packed particles, significantly increases surface area and increases surface reactivity. The presence of oxygen peaks in the EDS spectrum of the LS points to the formation of titanium oxide (TiO_2_), a result of the laser modification. This oxide layer is known to improve surface properties, including wettability and biocompatibility, making the surface more conducive to bone cell attachment and proliferation. Such modifications were observed in a previous study, indicating that increased surface roughness and the presence of bioactive compounds such as TiO_2_ improve osseointegration [46].

In LHS, a pattern similar to the LS surface was observed, since both had been modified by a laser. However, the deposition of HA on the surface created a highly porous and rough morphology, featuring numerous cavities and more irregular characteristics compared to the LS. The EDS spectrum for the LHS showed a complex composition with peaks of oxygen, calcium, phosphorus and other elements, confirming the successful deposition of HA. This bioactive coating increases the surface’s ability to interact with biological tissues, promoting better osseointegration and stability by mimicking natural bone composition [49,50,51]. The superhydrophilic nature of LSs and LHSs, demonstrated by their 0° contact angles, ensures the immediate and complete distribution of fluids, which is crucial for effective cellular interactions and bone formation. These characteristics are well-documented in the literature, where HA coatings are shown to improve bone–implant contact and integration [10,40].

The quantitative analysis of bone-to-implant contact (%BIC) further substantiated these findings. The percentage of BIC in mineralized tissues has been a “gold standard” methodology for evaluating peri-implant bone formation [46,52]. In the present study, it was possible to observe that the percentages of BIC in the LS (57.94 ± 10.04 and 60.50 ± 7.64 at 2 and 4 weeks, respectively) and LHS (62.56 ± 3.31 and 62.40 ± 7.58 at 2 and 4 weeks, respectively) were statistically superior to that in the MS (32.68 ± 7.22) within 2 weeks and the MS (38.67 ± 9.24) within 4 weeks. These findings corroborate the results found from the confocal laser microscopy, which evaluates, among other analyses, the rate of bone apposition (MAR). MAR is obtained through measurements performed on images with superimposition of calcein and alizarin fluorochromes [44]. In this sense, the MAR of the LHS was statistically superior to that of the MS, and that of the LS was statistically superior to the MS in the period of 4 weeks. These histological and histometric findings can and should be related to the contact angle, which can be used to assess the degree of surface wettability. This superior performance of the LS and LHS can be attributed to their improved surface characteristics, which facilitate better initial bone contact and integration. These results are consistent with studies indicating that increased surface roughness and hydrophilicity lead to higher BIC percentages and better osseointegration [10,11,12,13,27,40,41].

Regarding new bone area (%NBA), our study found notable differences in the 2-week period between analyzed surfaces. The MS group had the lowest NBA, averaging around 65%, which highlights the limited effectiveness of untreated surfaces in promoting rapid bone formation. On the other hand, the LSs achieved approximately 80% NBA, and the LHSs had the highest NBA, almost 85%, demonstrating the enhanced bone formation capabilities conferred by these modifications. These observations are in line with the notion that surface roughness and hydrophilicity contribute significantly to early bone formation [53,54,55]. Over the 4-week period, although the MS group showed some improvement in NBA to about 70%, it still lagged behind the treated surfaces. Both LS and LHS experienced a slight reduction in NBA compared to their 2-week values, stabilizing at around 75%. This reduction may indicate that the initial accelerated bone formation stabilized as the healing process continued. Although there was no statistically significant difference between the treated surfaces at 4 weeks, it is evident that the LS and LHS consistently outperformed the MS throughout the study period (Figure 9). A previously published paper [13] demonstrated that laser-treated implants, as well as laser followed by HA deposition, exhibited greater BIC and NBA values compared to implants with a machined surface over 30-, 60-, and 90-day periods, with no significant differences observed between the treated surfaces or the time periods evaluated. This plateau effect on the treated surfaces is supported by the literature which suggests that after a rapid initial phase of bone formation, the rate of new bone growth may plateau as remodeling processes begin to dominate [56].

Our findings have relevant clinical implications. The use of biomaterials in the coating of dental implants aims to facilitate and accelerate the bone healing process, thus favoring the process of osseointegration. In vivo studies showed that implants treated with laser followed by HA deposition obtained promising results, with higher values of reverse torque and a greater amount of newly formed bone in the initial periods when compared to machined implants and implants treated with sandblasting followed by double acid etching [57]. In addition, this study showed that the modification of the surface of the implant by laser with or without an HA coating without thermic treatment provided important physicochemical modifications, allowing a faster anabolic modeling and with better levels of bone–implant contact.

The limitations of this study must be highlighted. As much as the results of the present study show that the modifications performed on the LSs and LHSs provided important physicochemical changes, allowing faster anabolic modeling and with better levels of bone–implant contact, the behavior of the experimental surfaces LS and LHS throughout a bench time should be considered, since the MS’s bench time is already well documented in the literature [58,59].

The novelty of this model is due to our evaluation of bone healing in the initial period (2 weeks), as well as all the advantages of using high-frequency laser for surface modification and the favorable results presented in this study and in previous studies [9,10,11,12,13]. In addition, studies on the incorporation of biomaterials into implant surfaces are extremely relevant, given the number of biomaterials that are commercially available today and their beneficial effects on the formation of bone tissue in areas with small defects. This fact directs us towards future research, which should use the technology associated with in vitro studies to evaluate, under physiological conditions and within the initial phases of the chronology of peri-implant bone healing, the biological behavior of osteoblastic cells, when exposed to surfaces modified by laser with and without the deposition of biomaterials such as silica, HA and beta tricalcium phosphate.

## 5. Conclusions

In view of the results obtained, and considering the limitations of the study, it can be concluded that the experimental modifications on the surfaces of the LS and LHS implants provided important physicochemical modifications that favored the deposition of bone tissue on the surface of the implants, increasing the bone–implant contact and allowing higher removal torque values, a consequence of the acceleration of the initial phases of the peri-implant bone healing process.

## Figures and Tables

**Figure 1 biology-13-00533-f001:**
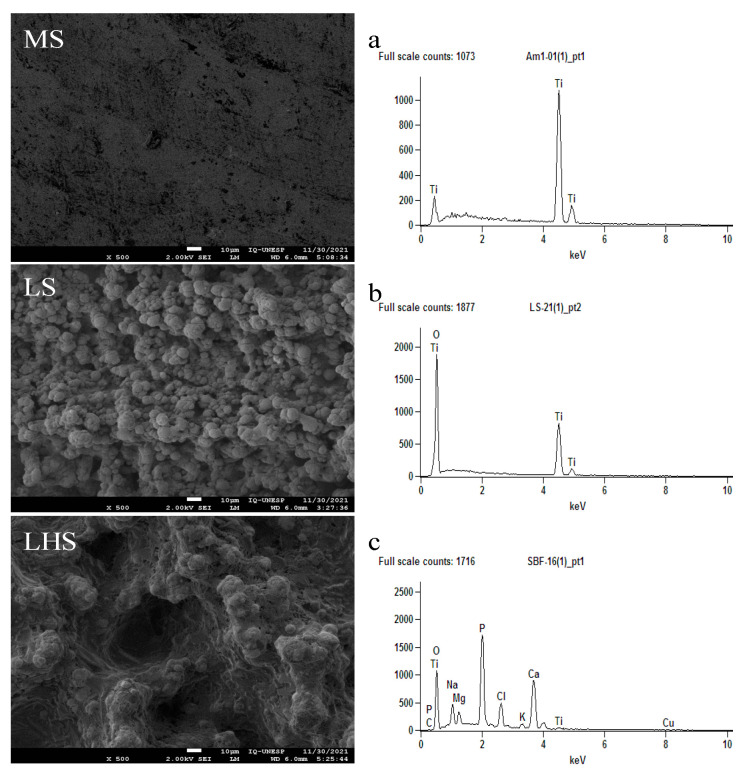
Scanning electron microscopy (SEM) and X-ray energy dispersive spectrometry (EDX) images offer a detailed analysis of the surface characteristics and chemical composition of the implants. This analysis includes MSs (**a**), LSs (**b**) and LHSs (**c**) before installation. The inset images offer a detailed view of the surface morphology at a high resolution, captured at 500× magnification. Scale bar is 10 μm.

**Figure 2 biology-13-00533-f002:**
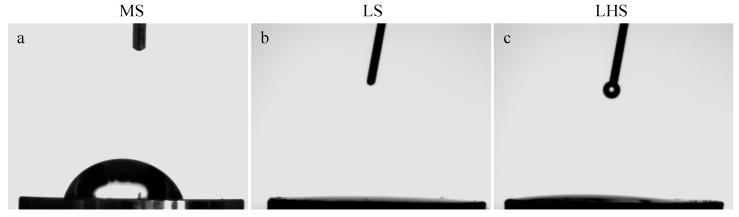
Representative surface contact angle image. MS (**a**), LS (**b**) and LHS (**c**).

**Figure 3 biology-13-00533-f003:**
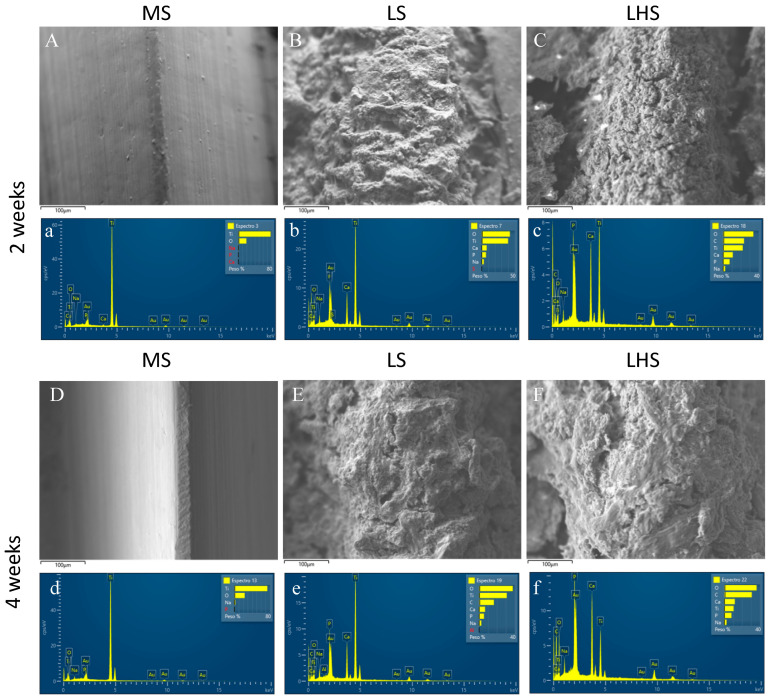
Scanning electron microscopy (SEM) and X-ray dispersive energy spectrometry (EDX) images illustrate the surface roughness patterns on implants removed by reverse torque. The surfaces analyzed include MS (**A**,**D**), LS (**B**,**E**), and LHS (**C**,**F**). The images reveal a greater presence of bone tissue on the LS and LHS at 2 and 4 weeks post-implantation. Additionally, EDX graphics show the elemental composition of the MS (**a**,**d**), LS (**b**,**e**), and LHS (**c**,**f**) surfaces at the 2 and 4-week intervals. The scale bar represents 100 µm.

**Figure 4 biology-13-00533-f004:**
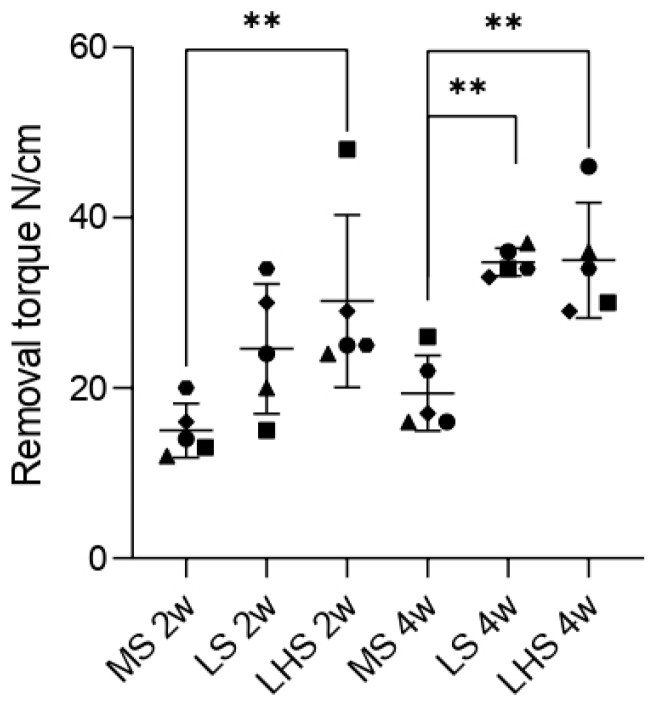
Removal torque measurements of the surfaces at 2 and 4 weeks. Data symbols represent individual measurements with median and standard deviation (SD). *p* < 0.05 in ANOVA (two way) and Tukey *t*-test represented by **.

**Figure 5 biology-13-00533-f005:**
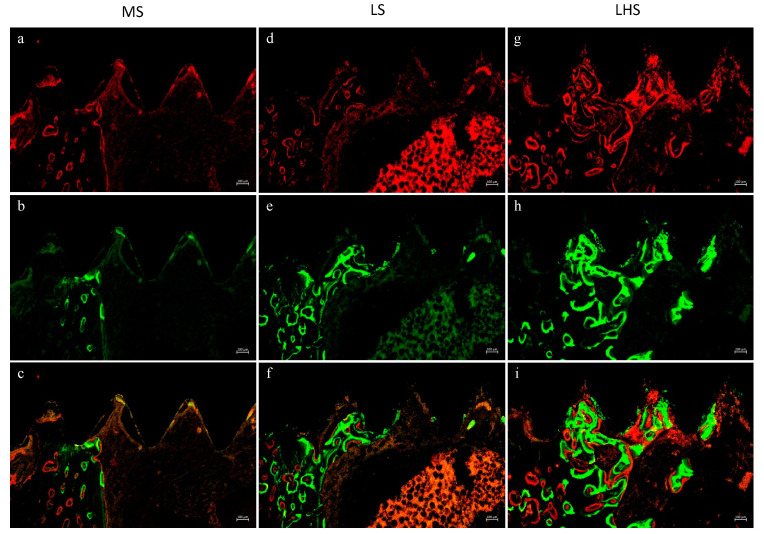
Images of precipitation of alizarin (red), calcein (green) and superposition of fluorochromes in MS (**a**–**c**); LS (**d**–**f**); and LHS (**g**–**i**). Fluorescence imaging demonstrated that MS showed moderate new bone formation and scattered remodeling activity. In contrast, LS and LHS exhibited more uniform and extensive new bone formation and active remodeling, suggesting superior performance in promoting bone growth and stability around dental implants. Scale bar is 100 µm.

**Figure 6 biology-13-00533-f006:**
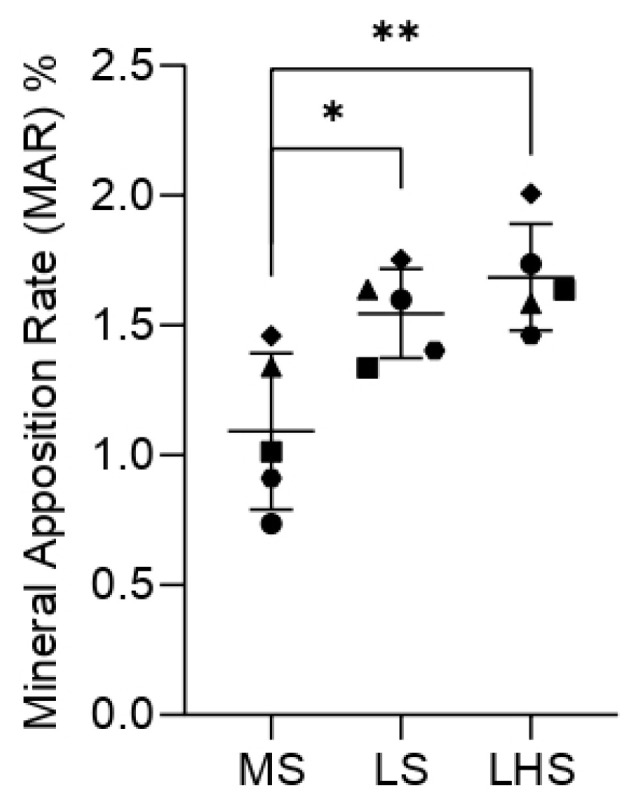
Mineral Apposition Rate (%MAR). Data points represent values from 5 individual measurements of the samples, shown with median and standard deviation (SD). Significant differences (*p* < 0.05) are indicated by * and **, determined using two-way ANOVA followed by Tukey’s *t*-test.

**Figure 7 biology-13-00533-f007:**
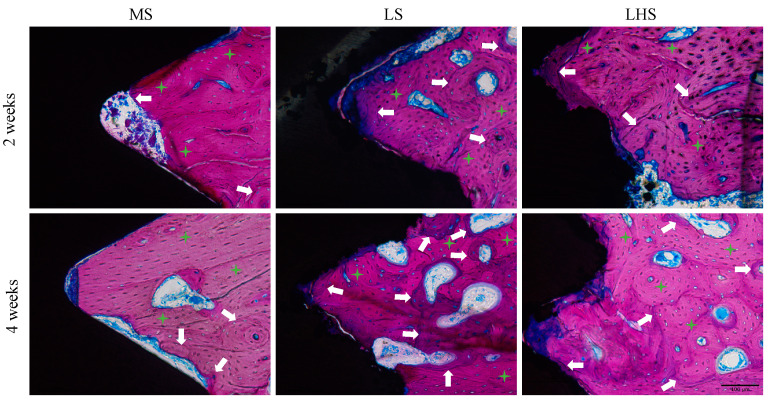
Histological sections of mineralized samples showing the upper cortical region, stained with Stevenel blue and acid fuchsin, at 200× magnification. White arrows indicate areas of bone matrix synthesis, demonstrating active bone formation, while green stars highlight regions of pre-existing bone without tissue remodeling, indicating areas where the bone has remained structurally unchanged. Scale bar is 100 μm.

**Figure 8 biology-13-00533-f008:**
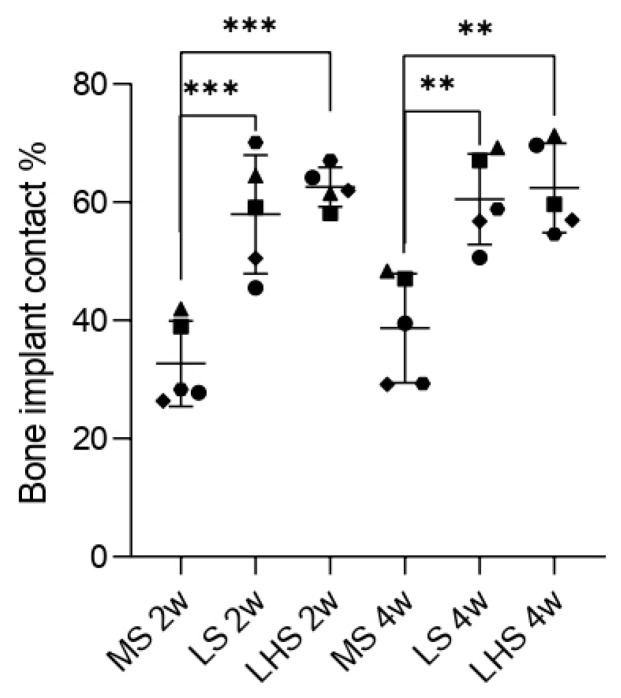
Bone implant contact (%BIC) at 2 and 4 weeks. Data points represent values from 5 individual measurements of the samples, shown with median and standard deviation (SD). Significant differences (*p* < 0.05) are indicated by ** and ***, determined using two-way ANOVA followed by Tukey’s *t*-test.

**Figure 9 biology-13-00533-f009:**
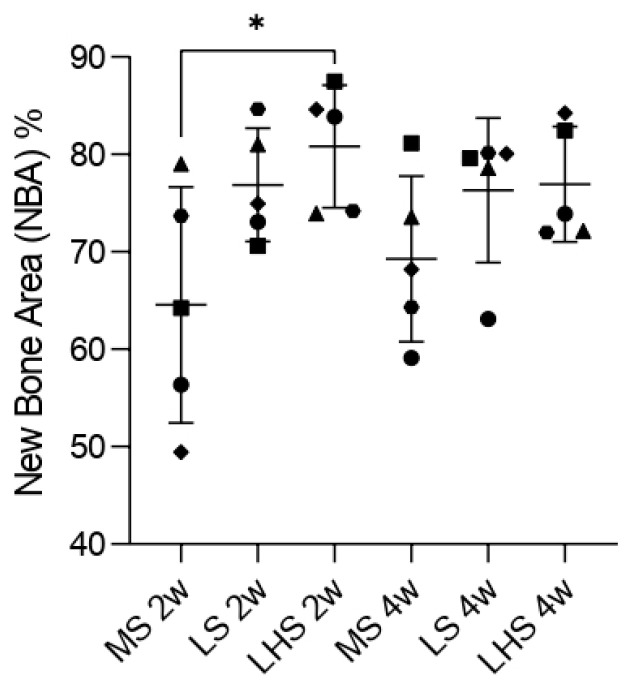
New bone area (%NBA) at 2 and 4 weeks. Data points represent values from 5 individual measurements of the samples, shown with median and standard deviation (SD). Significant difference (*p* < 0.05) is indicated by *, determined using two-way ANOVA followed by Tukey’s *t*-test.

**Table 1 biology-13-00533-t001:** Average value of the contact angle (θ) of the different surfaces.

Surface/Angle	MS	LS	LHS
First review	52.9° ± 9.58	0°	0°
Second review	57.1° ± 9.58	0°	0°
Third review	71.2° ± 9.58	0°	0°
Average	60.4° ± 5.53	0°	0°

Contact angle measurements indicating wettability characteristics of analyzed surfaces: LS and LHS demonstrate superior wettability compared to MS.

## Data Availability

A partial data presented in this study are openly available in UNESP Institutional Repository at http://hdl.handle.net/11449/244147 (accessed 15 June 2023) (Thesis).

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
