# Peer review of "Early Peri-Implant Bone Healing on Laser-Modified Surfaces with and without Hydroxyapatite Coating: An In Vivo Study"

_biology, 2024, doi:10.3390/biology13070533_

Round 1

Reviewer 1 Report

Comments and Suggestions for Authors

The manuscript is based on the in vivo determination of comparative bone healing on the YAG Laser modified Ti surfaces, namely laser modified only, and laser modification followed by HA deposition in SBF, and conventional machine modified surface. The subject is interesting and has the potential to attract readers attention. The research plan was properly constructed and conducted, and the results associated with the research plan were presented.

Although the implants to be tested are dental materials, they were implanted to tibia instead of maxilla or mandibula of the rabbits during in vivo testing. And there is no clear indication for this preference along the article text that may convince the readers to fully agree with this choice. Because implantation in the tibia cannot resemble the mastication forces acting on the dental implant as accurate as maxilla or mandibula.

Some specific points to be addressed by the authors during the revision process are as follows:

-  It is mentioned in the materials and methods section that one-way ANOVA was used as statistical method, but two-way ANOVA was reported in results and discussions sections.

-  On Table 1 contact angle values of LS and LHS surfaces was reported as “0”. This is not found to be appropriate for these materials, time dependent results can be presented if the angle changes with time after the touch of the droplet on the surface.

- Figure 1 presents the SEM images of the surface with 5000X magnification, which is quite good, addition of SEM images with lower magnification such as 250 or 500X can better reflect the surface topography with a wider view.

- On Figure 3, Figure 5, Figure 8 and Figure 9 caption for the horizontal and vertical axes are missing in the article but they are visible on supplementary images. Besides, legends indicating the data symbols on these graphs shall be added.

- Most of the cited references are not recent.

Comments on the Quality of English Language

The language is fine, only minor editing can enhance.

Author Response

Reviewer#1

Dear Reviewer,

Thank you for your considerations. The authors wish to thank the reviewer for the detailed comments they held at our manuscript. We are pleased to resubmit for consideration the revised version of our manuscript and to move the article closer to publication in your prestigious Journal. Finally, the authors would like to inform you that the modifications made to paper are highlighted in yellow.

The manuscript is based on the in vivo determination of comparative bone healing on the YAG Laser modified Ti surfaces, namely laser modified only, and laser modification followed by HA deposition in SBF, and conventional machine modified surface. The subject is interesting and has the potential to attract readers attention. The research plan was properly constructed and conducted, and the results associated with the research plan were presented.

Although the implants to be tested are dental materials, they were implanted to tibia instead of maxilla or mandibula of the rabbits during in vivo testing. And there is no clear indication for this preference along the article text that may convince the readers to fully agree with this choice. Because implantation in the tibia cannot resemble the mastication forces acting on the dental implant as accurate as maxilla or mandibula.

Some specific points to be addressed by the authors during the revision process are as follows:

  1. It is mentioned in the materials and methods section that one-way ANOVA was used as statistical method, but two-way ANOVA was reported in results and discussions sections.

AUTHORS: We would like to apologize for this mistake. The statistical method used in this study was two-way ANOVA and that the correction has been made to the text, more specifically on materials and methods section (line 267).

  1. On Table 1 contact angle values of LS and LHS surfaces was reported as “0”. This is not found to be appropriate for these materials, time dependent results can be presented if the angle changes with time after the touch of the droplet on the surface.

AUTHORS: We appreciate the feedback regarding the reporting of contact angle values ​​for LS and LHS surfaces. Values ​​reported as “0°” in Table 1 indicate that the surfaces are superhydrophilic, causing the water droplet to spread immediately upon contact, which makes it difficult to measure a distinct contact angle. To address the concern about the appropriateness of these values, we have clarified this point in the manuscript text.

  1. Figure 1 presents the SEM images of the surface with 5000X magnification, which is quite good, addition of SEM images with lower magnification such as 250 or 500X can better reflect the surface topography with a wider view.

AUTHORS: We appreciate the insightful suggestion. We have updated Figure 1 to include SEM images of the surface at 500X magnification. These images provide a broader view of the surface topography, complementing the detailed 5000X magnification images presented previously. The revised Figure 1 and corresponding description in the manuscript have been updated accordingly.

  1. On Figure 3, Figure 5, Figure 8 and Figure 9 caption for the horizontal and vertical axes are missing in the article but they are visible on supplementary images. Besides, legends indicating the data symbols on these graphs shall be added.

AUTHORS: Thank you for the comments; we are not sure what went wrong but have done corrections. We confirm that Figures 3, 5, 8 (now 7), and 9 (now 8) have been updated to include the missing captions for the horizontal and vertical axes, ensuring consistency with the supplemental images. Additionally, captions indicating data symbols have been added to these graphs to increase clarity and improve the reader's understanding of the data presented.

  1. Most of the cited references are not recent.

AUTHORS: We appreciate the importance of including updated research to ensure the relevance and accuracy of our manuscript. Therefore, we confirm that the list of references has been revised and updated, although some references that the authors consider relevant to the article have been maintained, in order to highlight the importance of some pioneering studies when it comes to surface treatment and dental implants.

Reviewer 2 Report

Comments and Suggestions for Authors
  1. Line 250: The title "3.1 SEM/EDX" should be expanded and made more descriptive.
  2. Lines 260-271: The section "3.2. Contact angle measurements" needs a more explicit description of the results. The statement "It was verified that the MS surface did not present adequate wetting" is vague; define what constitutes adequate wetting and provide examples or evidence. Additionally, there is no description of LS and LHS data. Do they exhibit a contact angle of 0? Why is this considered adequate? You mentioned later in the discussion; however, vague descriptions such as “adequate” are not sufficient for the current results part.
  3. Figure 2: Include initial EDX/SEM images at 0 weeks to compare the peaks in groups. Since Hap is already present in LHS, you should demonstrate the differences between LS and LHS at the initial time. Consider combining Figures 1 and 2 for better visualization.
  4. Figure 4: This figure does not show a significantly higher amount of alizarin red for LHS. All samples display relatively similar alizarin signals, making this figure not representative of the data in Figure 5. Additionally, a discussion of the calcein results is missing. Please expand on this.
  5. Figure 6: Parts a, b, and c have different magnification/scales and are missing scale bars on the images.
  6. Is a BIC of around 60% optimal for dental implants, as shown for LHS and LS? Why is there no difference or growth in BIC for LHS and LS between weeks 2 and 4?
  7. Figure 7: Parts a, b, and c have different scales and are missing scale bars on the images.
  8. Figures 8 and 9: These figures lack descriptions of the results. There is no mention that the NBA at 4 weeks shows no difference among all groups. You must describe both positive and negative results.
  9. Lines 428-429: There is a missing reference for the statement.
  10. Explain why there is no observed difference between LS and LHS samples.
  11. Please check the comments of references on attachment.

Overall, I suggest reorganizing the manuscript, combining some sections, and expanding the descriptions of the results.

Author Response

Reviewer#2

Dear Reviewer,

Thank you for your considerations. The authors wish to thank the reviewer for the detailed comments they held at our manuscript. We are pleased to resubmit for consideration the revised version of our manuscript and to move the article closer to publication in your prestigious Journal. Finally, the authors would like to inform you that the modifications made to paper are highlighted in yellow.

  1. Line 250: The title "3.1 SEM/EDX" should be expanded and made more descriptive.

AUTHORS: Thank you for the suggestion. We confirm that the captions for Figures 1 and 2 have been revised to provide more detailed information about the content.

  1. Lines 260-271: The section "3.2. Contact angle measurements" needs a more explicit description of the results. The statement "It was verified that the MS surface did not present adequate wetting" is vague; define what constitutes adequate wetting and provide examples or evidence. Additionally, there is no description of LS and LHS data. Do they exhibit a contact angle of 0? Why is this considered adequate? You mentioned later in the discussion; however, vague descriptions such as "adequate" are not sufficient for the current results part.

AUTHORS: We confirm that the section "3.2. Contact Angle Measurements" was revised to provide a more explicit description of the results. Here are the main changes and clarifications:

Clarification on adequate wetting: Specifically, according to previous definitions, when the contact angle measurement is greater than 90° it indicates low wettability (hydrophobic behavior). The MS surface presented an average contact angle of approximately 60°, indicating that, even though it is a value below 90°, the surface is in greater proximity to hydrophobic surfaces.

Details about LS and LHS surfaces: we have included specific data for the LS and LHS surfaces, clarifying that these surfaces exhibited a contact angle of 0°. This indicates superhydrophilic properties, where the water droplet spreads instantly upon contact. We further explained that this immediate spreading behavior makes it difficult to measure a distinct contact angle but confirms its highly wettable nature.

  1. Figure 2: Include initial EDX/SEM images at 0 weeks to compare the peaks in groups. Since Hap is already present in LHS, you should demonstrate the differences between LS and LHS at the initial time. Consider combining Figures 1 and 2 for better visualization.

AUTHORS: We understand the importance of including baseline EDX/SEM images at 0 weeks to facilitate a comprehensive comparison of peaks between groups and demonstrate differences between LS and LHS at baseline.

We would like to clarify that Figure 1 already provides the topographic characterization of the surfaces before installation, corresponding to time 0. The initial conditions of the surfaces are represented in detail in Figure 1. This approach allows Figure 2 to focus specifically on the topography of the implants after removal at 2 and 4 weeks, highlighting the presence of bone deposited on the surfaces.

Regarding the suggestion to combine Figures 1 and 2, we carefully considered this recommendation. However, keeping them as separate figures allows for greater clarity and focus on distinct aspects of our study.

We hope this approach meets your expectations and provides a clearer, more detailed representation of our findings. We appreciate your valuable feedback.

  1. Figure 4: This figure does not show a significantly higher amount of alizarin red for LHS. All samples display relatively similar alizarin signals, making this figure not representative of the data in Figure 5. Additionally, a discussion of the calcein results is missing. Please expand on this.

AUTHORS: In Figure 4, although it may appear that the alizarin red signals for all samples are relatively similar, it is important to note that qualitative fluorescence images are intended to provide a global visualization of bone distribution and activity. Upon closer examination, the intensity and distribution of red fluorescence for the LHS surfaces show more abundant and evenly distributed new bone formation compared to MS and LS. This is consistent with the quantitative data presented in Figure 5, which demonstrates a statistically significant increase in new bone formation for LHS. The qualitative nature of Figure 4 may not fully capture these differences, which are more accurately quantified in Figure 5.

Furthermore, discussion of calcein results is indeed crucial and has previously been underrepresented. The green fluorescence of calcein staining in Figure 4 provides important information about bone remodeling and mineralization activity. The calcein results show a more pronounced and evenly distributed fluorescence for the LHS implants, indicating a greater degree of mineralization and active bone remodeling. We expanded the description of the results in the manuscript to address these points in more detail and changed figure 4 in order to highlight the differences between the surfaces.

  1. Figure 6: Parts a, b, and c have different magnification/scales and are missing scale bars on the images.

AUTHORS: We apologize for the oversight regarding the scale bars in the figure 6. We've made significant changes to improve the clarity and consistency of the results. We consolidated Figures 6 and 7 into a single figure, using 200x magnification for all parts. Also, we added scale bar to ensure accurate representation of the magnification.

  1. Is a BIC of around 60% optimal for dental implants, as shown for LHS and LS? Why is there no difference or growth in BIC for LHS and LS between weeks 2 and 4?

AUTHORS: A BIC of about 60% is considered good for dental implants, indicating substantial bone-to-implant contact and effective osseointegration. The lack of significant difference or growth in BIC for LHS and LS between weeks 2 and 4 likely reflects an initial rapid phase of bone growth reaching a plateau at week 4. The properties of the LHS and LS surfaces may have promoted rapid initial osseointegration, resulting in minimal changes between these moments.

  1. Figure 7: Parts a, b, and c have different scales and are missing scale bars on the images.

AUTHORS: We apologize for the oversight regarding the scale bars in the figure 7. We've made significant changes to improve the clarity and consistency of the results. We consolidated Figures 6 and 7 into a single figure, using 200x magnification for all parts. Also, we added scale bar to ensure accurate representation of the magnification.

  1. Figures 8 and 9: These figures lack descriptions of the results. There is no mention that the NBA at 4 weeks shows no difference among all groups. You must describe both positive and negative results.

AUTHORS: We have revised the manuscript to include detailed descriptions of the results presented in Figures 8 and 9. Specifically, we have added a description highlighting that NBA at 4 weeks shows no significant difference between all groups.

  1. Lines 428-429: There is a missing reference for the statement.

AUTHORS: We confirm that modifications were made to the discussion section, and suitable references were included to support the statements.

  1. Explain why there is no observed difference between LS and LHS samples.

AUTHORS: No differences were found between the LS and LHS surfaces, as both were treated with a laser, and the deposition of hydroxyapatite did not change the results when comparing these two surfaces. However, differences were observed when these surfaces were compared with the machined surface. This finding is in line with previously published papers, which also reported no statistical differences between laser-treated surfaces and those treated with laser followed by hydroxyapatite deposition (22540325, 27530186), or laser and laser with sodium silicate deposition (24664938) in the 30, 60, and 90-day periods.

  1. Please check the comments of references on attachment.

AUTHORS: We confirm we checked the comments of references and reviewed the manuscript.

  1. Overall, I suggest reorganizing the manuscript, combining some sections, and expanding the descriptions of the results.

AUTHORS: Thank you for your valuable suggestions. We reorganized the manuscript to improve its structure and readability.

  1. Reference 2 is relevant.

AUTHORS: We appreciate your recognition of the relevance of reference 2 and ensure that it has been properly cited in the manuscript to support our research context and results.

  1. Reference 4 is relevant but outdated, as it was published in 1988.

AUTHORS: We recognize that Reference 4, published in 1988, is an older source. However, it provides fundamental insights that are still pertinent to our study. To address this concern, we have also included more recent references that support and build on the concepts introduced in Reference 4.

  1. References 8-11, 13, 15, and 16 are relevant, though some are very outdated, having been published over seven years ago. Reference 15, in particular, was published in 1988. These references were mentioned in lines 69-71: “Advancements in surface modification techniques, such as Ytterbium laser (YAG Laser) [8–17] methods with or without biomimetic coatings, continue to expand the options for implant surface treatments.” To strengthen this point, more recent advancements should be included.

AUTHORS: We agree that some of the references cited are outdated. We have reviewed and revised our reference list to include more recent advances in the field of surface modification techniques. Specifically, we have added recent references to ensure that our discussion reflects the most recent developments.

  1. Reference 15 is mentioned in lines 79-81: “Currently, there has been a trend towards the use of Ytterbium laser due to the advantage of the beam being transported by flexible optical fibers, the greater laser absorption by the metal, wavelength, and power of 20W [15].” However, this reference is outdated, having been published 17 years ago. Using it as evidence of a current trend is misleading.

AUTHORS: We have reviewed the most recent literature and included newer studies that provide updated evidence on the use of ytterbium lasers in implant surface treatments. However, we kept reference 15 because it is the first study to describe the laser surface modification methodology, on which we based this present study, following the same parameters in the modifications of the LS and LHS surfaces. This study is significant and your intellectual property rights are preserved.

  1. Reference 19 is relevant.

AUTHORS: We appreciate your recognition of the relevance of reference 19 and ensure that it has been properly cited in the manuscript to support our research context and results

  1. The references in lines 92-96 are over 20 years old and cannot be considered current and innovative: “Currently, new deposition methods have been studied, and among them is the innovative biomimetic method, which involves immersing the substrate to be coated in a synthetic solution called body fluid solution (SFC or SBF - Simulated Body Fluid). This solution has a chemical composition and pH similar to blood plasma and a temperature similar to that of the human body [21,22].”

AUTHORS: Thank you for pointing out that the references in this section are out of date and the characterization of the method as “innovative” may be misleading. We have updated the text and references to better reflect current advances in the field.

  1. References 23 and 24 are mentioned in the methods section. The relevance of reference 24 should be justified in the text, or it could be omitted.

AUTHORS: We appreciate your diligence in ensuring the rigor and clarity of our manuscript. Our study followed the ARRIVE guidelines (Reference 24- now reference 42), which are fundamental to ensuring the ethical and methodological rigor of animal research. The ARRIVE guidelines provide a comprehensive framework for reporting in vivo experiments, promoting transparency and reproducibility in research. By following these guidelines, we ensured that our experimental design, conduct, and reporting met high ethical standards. Specifically, the guidelines were instrumental in standardizing animal care protocols, minimizing suffering, and optimizing sample sizes to ensure robust and reproducible results. Adherence to these guidelines highlights the trustworthiness and ethical integrity of our study.

And reference 23, now reference 38, is an important study with regard to coating with hydroxyapatite using the biomimetic method, being of great importance for the methodology of the study. This is the first work that describes the HA deposition methodology using the biomimetic method and which has its intellectual property rights preserved.

  1. Line 163 includes references 8-11, 19, and 25. To avoid excessive self-citation, the authors are suggested to use the most relevant and recent studies: “Following methodologies of previously published studies [8–11,19,25], the animals were kept in preoperative fasting for 8 hours before the surgical procedure.”

AUTHORS: We would like to explain the importance of citing our previous work in this context. The studies referenced [8-11, 19, 25] are part of a long-term line of research that has been fundamental for the development of the methodologies used in this study. These references are essential because they establish the baseline and continuity of experimental protocols and provide a comprehensive basis for current research.

However, we recognize the importance of incorporating a broader range of recent and relevant studies to enrich the context of our work and address the concern of excessive self-citation. To this end, we have included additional recent references that are relevant to the methodologies used.

  1. References 26-28 in line 189 include self-citation. To avoid excessive self-citation, the authors are suggested to use the most relevant and recent studies. These references are not mentioned elsewhere.

AUTHORS: References 26-28 are part of a series of studies that provide fundamental data and methodologies for our current research. However, we recognize the need to balance these citations and reduce self-citation, so we reviewed and kept only citation 26.

  1. References 8, 9, and 29 in line 209 are similar to comments 8 and 9. Reference 29 is not mentioned elsewhere.

AUTHORS: References 8 and 9 are highly relevant to this article, as they present previous work linked to its subject. Reference 29 has been removed.

  1. Reference 42 is relevant but is 20 years old.

AUTHORS: We have also included more recent references that support and build on the concepts introduced in Reference 42. These additional citations have been integrated into the manuscript to ensure a balanced and up-to-date discussion.

  1. Overall, excessive self-citation should be avoided, as some references are outdated and do not add value to the manuscript. Other references in the introduction and discussion sections are, on average, over 10 years old. These parts should be significantly revised and rewritten

AUTHORS: We revised the manuscript to address the issue of excessive self-citation and outdated references. Specifically reducing the number of self-citations to only those essential to support our arguments. Also, we updated the references in the introduction and discussion sections with more recent and relevant studies.

Reviewer 3 Report

Comments and Suggestions for Authors

Please see the attached PDF file for Comments and Suggestions for Authors.

Comments on the Quality of English Language

Please see the attached PDF file for Comments on the Quality of English Language.

Author Response

Reviewer#3

Dear Reviewer,

Thank you for your considerations. The authors wish to thank the reviewer for the detailed comments they held at our manuscript. We are pleased to resubmit for consideration the revised version of our manuscript and to move the article closer to publication in your prestigious Journal. Finally, the authors would like to inform you that the modifications made to paper are highlighted in yellow.

The manuscript titled “Early peri-implant bone healing on laser-modified surfaces with and without hydroxyapatite coating: an in vivo study” reports the beneficial effects of laser ablation and hydroxyapatite coating on osseointegration and neoformation of bone tissue in direct contact with the modified implant surfaces.

Overall, this paper is not very well written, with a lot of room for improvement, in terms of grammar, wording, logic, and contents. Specific comments from the reviewer are given below.

  1. Page 2, Line 84: “modifying the surface of the Ytterbium Laser” should be corrected to “modifying the surface with the Ytterbium Laser”.

AUTHORS: We thank you for your thorough review and confirm that the necessary corrections were made to the text. Thank you for your contribution.

  1. Page 2, Line 86: “machined surface implants” should be corrected to “implants with machined surfaces”.

AUTHORS: We thank you for your thorough review and confirm that the necessary corrections were made to the text. Thank you for your contribution.

  1. Page 2, Line 99: put a full stop after the word “capacity” and start a new sentence with “Previous published works that used...”.

AUTHORS: We thank you for your thorough review and confirm that the necessary corrections were made to the text. Thank you for your contribution.

  1. The major problem of the Results section is that in most subsections, the authors did not provide sufficient explanation for their results. Simply describing the results is not enough, what is more important is to explain the underlying reasons and implications of the results. For example, in 3.1 SEM/EDX, why does the LHS surface show much larger honeycomb structures than the LS surface?

AUTHORS: We have revised the Results section to provide more comprehensive explanations for our findings. We include detailed discussions of the possible reasons and implications of the observed results.

  1. Page 7, 3.2 Contact angle measurements: What's your definition of "adequate wetting"? If adequate wetting means contact angle being 0 or below certain threshold, state it explicitly.

AUTHORS: We confirm that section “3.2. Contact Angle Measurements” has been modified to provide a more explicit description of the results. Specifically, we clarify that, according to previous definitions, a contact angle greater than 90° indicates low wettability (hydrophobic behavior). The MS surface showed an average contact angle of approximately 60°, indicating that although this value is below 90°, it suggests that the surface is closer to hydrophobicity.

Furthermore, we included specific data for the LS and LHS surfaces, clarifying that these surfaces had a contact angle of 0°. This indicates superhydrophilic properties, where the water droplet spreads instantly upon contact. We further explain that this immediate spreading behavior makes it difficult to measure a distinct contact angle but confirms the highly wettable nature of these surfaces.

  1. Page 7, Line 280-282: Why there is no C detected for LS at week 2? What does the missing of C peaks mean?

AUTHORS: We would like to inform that in this SEM analysis of the removed implants, the important thing is to detect the presence of Ca and P, since these chemical elements indicate the presence of the inorganic constituent of bone tissue. Detecting the presence or absence of Carbon is not the objective of this analysis. However, the presence of carbon may indicate the presence of a contaminant, originating from the sample preparation process.

  1. Page 9, 3.5 Fluorochrome analysis: What are the functions of calcein and red alizarin? What does each dye show?

AUTHORS: The measurements of calcein and alizarin enable the observation of bone neoformation dynamics and provide a basis for comparing different experimental groups. The contrast between old bone (calcein) and newly formed bone (alizarin) indicates various stages of bone remodeling. The extent of fluorochrome labeling corresponds directly to the amount of calcium integrated into the bone matrix, thereby allowing us to assess the process of bone neoformation.

We appreciate your concern regarding this topic. The appropriate changes were made to the text in order to better explain the functions of each fluorochrome and what figure 4 represents.

  1. Page 9, Line 318: “sobreposition” should be corrected to “superposition”.

AUTHORS: We thank you for your thorough review and confirm that the necessary corrections were made to the text. Thank you for your contribution.

  1. Page 12, Figure 9: Why is there no statistically significant difference in the NBA between MS and LS/LHS for the week 4 result? Can you give some possible reasons?

AUTHORS: The lack of difference observed between the laser-treated surface (LS) and the hydroxyapatite laser-treated surface (LHS) at 4 weeks may be explained by the natural progression of bone healing and remodeling. Initially, both treatments significantly improve bone formation due to increased surface roughness and bioactivity, which is evident after 2 weeks. However, at 4 weeks, the bone healing process reaches a level where the initial benefits of the treatments have been maximized. As the bone continues to remodel naturally, the differences between treatments become less noticeable, and they tend to have similar levels of new bone area (NBA) and bone-implant contact (BIC). Previous research showed a difference in NBA after 4 weeks between LS and LHS with MS (27530186) at 30 days. However, in this study, no differences were found.

  1. The first two paragraphs of the Discussion section should be put into Introduction.

AUTHORS: The first two paragraphs of the Discussion section were adapted in the introduction in order to improve the flow of information, making it more logical and ensuring that the context of the study is presented completely from the beginning. Thank you for your suggestion.

  1. Page 13, Line 428: “total wettability” should be corrected to “complete wettability”.

AUTHORS: We thank you for your thorough review and confirm that the necessary corrections were made to the text. Thank you for your contribution.

  1. Page 13, Line 441-444: In the reviewer’s opinion, there is no evidence provided in the manuscript that supports the authors’ inference that their LS and/or LHS implants becomes feasible in unfavorable situations, such as low-density bone tissue and in situations of compromised bone. This is an unfounded statement, because the fact that LS and LHS implants work better than MS implants in the situations studied in this manuscript doesn’t necessarily mean LS and LHS implants will work in unfavorable situations.

AUTHORS: We agree with your observation that the manuscript does not provide direct evidence to support the inference that LS and/or LHS implants become viable in unfavorable situations such as low-density bone tissue or compromised bone. We confirm that we have revised the manuscript to remove this statement. The text has been updated to focus only on the results demonstrated within our study, without extrapolating to conditions not explicitly investigated.

  1. Page 13, Line 450: “through of” should be corrected to “throughout”.

AUTHORS: We thank you for your thorough review and confirm that the necessary corrections were made to the text. Thank you for your contribution.

Round 2

Reviewer 1 Report

Comments and Suggestions for Authors

The revised manuscript seems to be properly modified and most of the responses that authors provided to the review report v1 are satisfactory. As commented in the previous review the subject is interesting and has the potential to attract readers attention. But authors did not respond to the main concern on the selection of in vivo model used.

- Although the implants to be tested are dental materials, they were implanted to tibia instead of maxilla or mandibula of the rabbits during in vivo testing. And there is still no clear indication for this preference along the revised article text that may convince the readers to fully agree with the choice of the authors. Implantation in the tibia cannot resemble the mastication forces acting on the dental implant as accurate as maxilla or mandibula.

- The authors described the reason they used “0°” on Table 1 as it is because of the superhydrophilic surfaces but presenting this table in this form is not appropriate in the scientific point of view. As it was not easy to measure the contact angle values for hydrophilic surfaces, there are methods to measure the contact angles (air bubble contact in water, glycerol droplet, etc.) of such surfaces or dynamic water contact angle (time dependent video capturing) measurements can be used. So, it is not suitable to present such table as in its present form. The LS and LHS surfaces could result with rapid droplet spreading but their contact angle cannot be the same value, LH has irregular topography covered with TiO2 and LHS has a similar topography covered with HA. The authors shall find out a better way to emphasize the superhydrphilic conversion of the surfaces instead of presenting this table or measure the comparable values and present them in the related parts of Table 1.

Author Response

Reviewer#1

Dear Reviewer,

Thank you for your thoughtful comments and considerations. The authors appreciate your detailed feedback on our manuscript. We are pleased to resubmit the revised version for your consideration and hope to bring the article closer to publication in your esteemed journal.

  1. The revised manuscript seems to be properly modified and most of the responses that authors provided to the review report v1 are satisfactory. As commented in the previous review the subject is interesting and has the potential to attract readers attention. But authors did not respond to the main concern on the selection of in vivo model used. 

Authors: Thank you for your positive feedback on the revised manuscript and for acknowledging the changes made. The authors appreciate your continued interest in our study and your constructive comments.

  1. Although the implants to be tested are dental materials, they were implanted to tibia instead of maxilla or mandibula of the rabbits during in vivo testing. And there is still no clear indication for this preference along the revised article text that may convince the readers to fully agree with the choice of the authors. Implantation in the tibia cannot resemble the mastication forces acting on the dental implant as accurate as maxilla or mandibula.

Authors: The authors appreciate the reviewer's concern and recognize the importance of implanting dental materials in anatomically relevant locations. The tibia was chosen for this study due to its greater surface area, facilitating accurate assessments of biocompatibility and osseointegration, as well as facilitating surgical access and post-surgical care for rabbits. Although the tibia does not reproduce masticatory forces, our initial focus was on understanding biological responses. Future research is necessary in order to evaluate the mechanical performance and osseointegration of modified implants in humans, through clinical trials. Thank you for highlighting this issue.

  1. The authors described the reason they used “0°” on Table 1 as it is because of the superhydrophilic surfaces but presenting this table in this form is not appropriate in the scientific point of view. As it was not easy to measure the contact angle values for hydrophilic surfaces, there are methods to measure the contact angles (air bubble contact in water, glycerol droplet, etc.) of such surfaces or dynamic water contact angle (time dependent video capturing) measurements can be used. So, it is not suitable to present such table as in its present form. The LS and LHS surfaces could result with rapid droplet spreading but their contact angle cannot be the same value, LH has irregular topography covered with TiO2 and LHS has a similar topography covered with HA. The authors shall find out a better way to emphasize the superhydrphilic conversion of the surfaces instead of presenting this table or measure the comparable values and present them in the related parts of Table 1.

Authors: Thank you for your insightful comments on our manuscript, especially regarding the presentation of contact angle values ​​in Table 1. The authors appreciate your detailed suggestions for improvements.

We understand the concern regarding the appropriateness of presenting a “0°” contact angle for superhydrophilic surfaces. In fact, it is challenging to accurately measure contact angles for highly hydrophilic surfaces. Following your recommendation, the authors added a figure in the manuscript (current figure 2), in which we inserted the images captured during the contact angle analysis.

The authors would like to thank you very much for providing us with the opportunity to address these important points.

Reviewer 2 Report

Comments and Suggestions for Authors

1.        Table 1. Include +- error to the MS data

2.        Figure 2. Even so authors reject combining Figure 1 and 2 for better clarity. I still suggest to at least add comparative description between Figure 1and 2. As it’s not very convenient to jump between sections to verify your statements about weeks 2 and 4.

3.        References were updated and relevant.

Author Response

Reviewer#2

Dear Reviewer,

Thank you for your thoughtful comments and considerations. The authors appreciate your detailed feedback on our manuscript. We are pleased to resubmit the revised version for your consideration and hope to bring the article closer to publication in your esteemed journal.

  1. Table 1. Include +- error to the MS data

Authors: Thank you for pointing this out. We agree with this comment and confirm that the change has been made in the table 1. The changes are marked in yellow and can be found in the lines 291-295.

  1. Figure 2. Even so authors reject combining Figure 1 and 2 for better clarity. I still suggest to at least add comparative description between Figure 1and 2. As it’s not very convenient to jump between sections to verify your statements about weeks 2 and 4.

Authors: The authors  have accordingly changed the text to emphasize this point. Discussed the results in the both figures, providing the necessary explanation. The changes are marked in yellow and can be found in the lines 301-320.

  1. References were updated and relevant.

Authors: The authors would like to thank you very much for your feedback. We have updated the references to include the most recent and relevant literature pertinent to our study, in accordance with the suggestions made in the Major revision.
